# A Bayesian Approach to Generative Adversarial Imitation Learning

**Wonseok Jeon**[1], **Seokin Seo**[1], **Kee-Eung Kim**[1,2]
[1] School of Computing, KAIST, Republic of Korea
[2] PROWLER.io
{wsjeon, siseo}@ai.kaist.ac.kr, kekim@cs.kaist.ac.kr

## Abstract

Generative adversarial training for imitation learning has shown promising results on high-dimensional and continuous control tasks. This paradigm is based on reducing the imitation learning problem to the density matching problem, where the agent iteratively refines the policy to match the empirical state-action visitation frequency of the expert demonstration. Although this approach can robustly learn to imitate even with scarce demonstration, one must still address the inherent challenge that collecting trajectory samples in each iteration is a costly operation. To address this issue, we first propose a Bayesian formulation of generative adversarial imitation learning (GAIL), where the imitation policy and the cost function are represented as stochastic neural networks. Then, we show that we can significantly enhance the sample efficiency of GAIL leveraging the predictive density of the cost, on an extensive set of imitation learning tasks with high-dimensional states and actions.

## 1  Introduction

Imitation learning is the problem where an agent learns to mimic the demonstration provided by the expert, in an environment with unknown cost function. Imitation learning with policy gradients [Ho et al., 2016] is a recently proposed approach that uses gradient-based stochastic optimizers. Along with trust-region policy optimization (TRPO) [Schulman et al., 2015] as the optimizer, it is shown to be one of the most practical approaches that scales well to large-scale environments, i.e. high-dimensional state and action spaces. Generative adversarial imitation learning (GAIL) [Ho and Ermon, 2016], which is of our primary interest, is a recent instance of imitation learning algorithms with policy gradients. GAIL reformulates the imitation learning problem as a density matching problem, and makes use of generative adversarial networks (GANs) [Goodfellow et al., 2014]. This is achieved by generalizing the representation of the underlying cost function using neural networks, instead of restricting it to the class of linear functions for the sake of simpler optimization. As a result, the policy being learned becomes the generator, and the cost function becomes the discriminator. Based on the promising results from GAIL, a number of improvements appeared in the literature [Wang et al., 2017, Li et al., 2017].

Yet, one of the fundamental challenges lies in the fact that obtaining trajectory samples from the environment is often very costly, e.g., physical robots situated in real-world. Among a number of improved variants of GAIL, we remark that generative moment matching imitation learning (GM-MIL) [Kim and Park, 2018], which uses kernel mean embedding to improve the discriminator training just as in generative moment matching networks (GMMNs) [Li et al., 2015], was experimentally shown to converge much faster and more stable compared to GAIL. This gives us a hint that a robust discriminator is an important factor in improving the sample efficiency of generative-adversarial approaches to imitation learning.

In this work, we also aim to enhance the sample efficiency of the generative-adversarial approach to imitation learning. Our main idea is to use a Bayesian discriminator in GAIL, e.g. using a Bayesian neural network, thus referring to our algorithm as Bayes-GAIL (BGAIL). To achieve this, we first reformulate GAIL in the Bayesian framework. As a result, we show that GAIL can be seen as optimizing a surrogate objective in our approach, with iterative updates being maximum-likelihood (ML) point estimations. In our work, instead of using the ML point estimate, we propose to use the predictive density of the cost. This gives more informative cost signals for the policy training and makes BGAIL significantly more sample-efficient compared to the original GAIL.

## 2 Preliminaries

### 2.1 Reinforcement Learning (RL) and Notations

We first define the basic notions from RL. The RL problem considers an agent that chooses an action after observing an environment state and the environment that reacts with a cost and a successor state to the agent's action. The agent-environment interaction is modeled by using a Markov decision process (MDP) $\mathcal{M} := \langle S, A, c, P_T, \nu, \gamma \rangle$; $S$ is a state space, $A$ is an action space, $c(s, a)$ is a cost function, $P_T(s'|s, a)$ is the state transition distribution, $\nu(s)$ is the initial state distribution, $\gamma \in [0, 1]$ is a discount factor. $\mathcal{M}^-$ denotes an MDP $\mathcal{M}$ without the cost function (MDP\C), i.e., $\langle S, A, P, \nu, \gamma \rangle$. The (stochastic) policy $\pi(a|s)$ is defined as the probability of choosing action $a$ in state $s$.

Given the cost function $c$, the objective of RL is to find the policy $\pi$ that minimizes the expected long-term cost $\eta(\pi, c) := \mathbb{E}_\pi \left[ \sum_{t=0}^\infty \gamma^t c(\boldsymbol{s}_t, \boldsymbol{a}_t) \right]$, where the subscript $\pi$ in the expectation implies that the trajectory $(\boldsymbol{s}_0, \boldsymbol{a}_0, \boldsymbol{s}_1, \boldsymbol{a}_1, ...)$ is generated from the policy $\pi$ with the transition distribution of $\mathcal{M}^-$. The state value function $V_\pi^c$ and the action value function $Q_\pi^c$ are defined as $V_\pi^c(s) := \mathbb{E}_\pi \left[ \sum_{t=0}^\infty c(\boldsymbol{s}_t, \boldsymbol{a}_t) | \boldsymbol{s}_0 = s \right]$ and $Q_\pi^c(s, a) := \mathbb{E}_\pi \left[ \sum_{t=0}^\infty c(\boldsymbol{s}_t, \boldsymbol{a}_t) | \boldsymbol{s}_0 = s, \boldsymbol{a}_0 = a \right]$, respectively. The optimal value functions $V_*^c, Q_*^c$ for $c$ are the value functions for the optimal policy $\pi_*^c := \arg\min_\pi \eta(\pi, c)$ under the cost function $c$. The $\gamma$-discounted state visitation occupancy $\rho_\pi$ for policy $\pi$ is defined as $\rho_\pi(s) := \mathbb{E}_\pi \left[ \sum_{t=0}^\infty \gamma^t \delta(s - \boldsymbol{s}_t) \right]$ for Dirac delta function $\delta$ when the state space $S$ is assumed to be continuous. For convenience, we denote the $\gamma$-discounted state-action visitation occupancy for $\pi$ as $\rho_\pi(s, a) := \rho_\pi(s)\pi(a|s)$. It can be simply shown that $\eta(\pi, c) = \mathbb{E}_{(\boldsymbol{s}, \boldsymbol{a}) \sim \rho_\pi}[c(\boldsymbol{s}, \boldsymbol{a})] := \sum_{s,a} \rho_\pi(s, a)c(s, a)$. Throughout this paper, bold-math letters are used to indicate random variables, and their realizations are written as non-bold letters.

### 2.2 Imitation Learning

Historically, behavioral cloning (BC) [Pomerleau, 1991] is one of the simplest approach to imitation learning, which learns to map the states to demonstrated actions using supervised learning. However, BC is susceptible to compounding error, which refers to small prediction error accumulated over time to a catastrophic level [Bagnell, 2015]. Inverse reinforcement learning (IRL) [Russell, 1998, Ng and Russell, 2000, Ziebart et al., 2008] is a more modern approach, where the objective is to learn the underlying unknown cost function that makes the expert optimal. Although this is a more principled approach to imitation learning, IRL algorithms usually involve planning as an inner loop, which usually requires the knowledge of transition distribution and mainly increases the computational complexity of IRL. In addition, IRL is fundamentally an ill-posed problem, i.e., there exist infinitely many cost functions that can describe identical policies, and thus requires some form of preferences on the choice of cost functions [Ng and Russell, 2000]. The Bayesian approach to IRL [Ramachandran and Amir, 2007, Choi and Kim, 2011] is one way of encoding the cost function preferences, which will be introduced in the following section.

Finally, imitation learning with policy gradients [Ho et al., 2016] is one of the most recent approaches, which replaces the costly planning inner loop with the policy gradient update in RL, making the algorithm practical and scalable. Generative adversarial imitation learning (GAIL) [Ho and Ermon, 2016] is an instance of this approach, based on the adversarial training objective

$$\arg\min_\pi \max_{D \in (0,1)^{S \times A}} \left\{ \mathbb{E}_{(\boldsymbol{s}, \boldsymbol{a}) \sim \rho_{\pi_E}}[\log D(\boldsymbol{s}, \boldsymbol{a})] + \mathbb{E}_{(\boldsymbol{s}, \boldsymbol{a}) \sim \rho_\pi}[\log(1 - D(\boldsymbol{s}, \boldsymbol{a}))] \right\}, \quad (1)$$

for a set $(0, 1)^{S \times A}$ of functions $D : S \times A \to (0, 1)$. This is essentially the training objective of GAN, where the generator is the policy $\pi$, and the discriminator $D$ is the intermediate cost function to be used in policy gradient update to match $\rho_\pi$ to $\rho_{\pi_E}$.

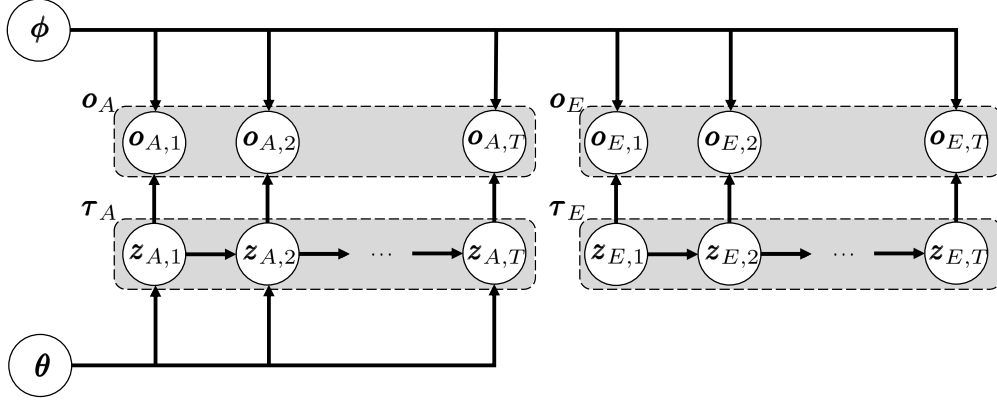

Figure 1: Graphical model for GAIL. The state-action pairs are denoted by $z := (s, a)$. Note that $p(z_1) = \nu(s_1)\pi_\theta(a_1|s_1)$ and $p(z_{t+1}|z_t) = \pi_\theta(a_{t+1}|s_{t+1})P_T(s_{t+1}|s_t, a_t)$. Also, the discriminator parameter $\phi$ and the policy parameter $\theta$ are regarded as random variables.

### 2.3 Bayesian Inverse Reinforcement Learning (BIRL)

The Bayesian framework for IRL was proposed by Ramachandran and Amir [2007], where the cost function $c$ is regarded as a random function. For the expert demonstration set $\mathcal{D} := \{\tau_n = (s_t^{(n)}, a_t^{(n)})_{t=1}^{T^{(n)}} | n = 1, ..., N\}$ collected under $\mathcal{M}^-$, the cost function preference and the optimality confidence on the expert's trajectories $\mathcal{D}$ are encoded as prior $p(c)$ and likelihood $p(\mathcal{D}|c)$, respectively. As for the likelihood, the samples in $\mathcal{D}$ are assumed independent Gibbs distribution with potential function $Q_*^c$, i.e. $p(\mathcal{D}|c) := \prod_{n=1}^{N} \prod_{t=1}^{T^{(n)}} p(a_t^{(n)}|s_t^{(n)}, c)$ for $p(a_t^{(n)}|s_t^{(n)}, c) \propto \exp(Q_*^c(s_t^{(n)}, a_t^{(n)})/\beta)$ with the temperature parameter $\beta$. Under this model, reward inference and imitation learning using the posterior mean reward were suggested. Choi and Kim [2011] suggested a BIRL approach using maximum a posterior (MAP) inference. Based on the reward optimality region [Ng and Russell, 2000], the authors found that there are cases where the posterior mean reward exists outside the optimality region, whereas MAP reward is posed inside the region. In addition, it was shown that the existing works on IRL [Ng and Russell, 2000, Ratliff et al., 2006, Syed et al., 2008, Neu and Szepesvári, 2007, Ziebart et al., 2008] can be viewed as special cases of MAP inference if we choose the likelihood and a prior properly.

## 3 Bayesian Generative Adversarial Imitation Learning

In order to formally present our approach, let us denote the agent's policy as $\pi_A$ and the expert's policy as $\pi_E$. In addition, let us denote sets $\mathcal{D}_A$ and $\mathcal{D}_E$ of trajectories generated by $\pi_A$ and $\pi_E$, respectively, under $\mathcal{M}^-$ for

$$\mathcal{D}_A := \left\{ \tau_A^{(n)} = (s_{A,t}^{(n)}, a_{A,t}^{(n)})_{t=1}^{T} \middle| n = 1, ..., N_A \right\}, \tag{2}$$

where the quantities for expert are defined in a similar way. In the remainder of this work, we drop the subscripts $A$ and $E$ if there is no confusion. Also, note that $\mathcal{D}_E$ will be given as input to the imitation learning algorithm, whereas $\mathcal{D}_A$ will be generated in each iteration of optimization. It is natural to assume that the agent's and the expert's trajectories $\tau_A$ and $\tau_E$ are independently generated, i.e., $p(\tau_A, \tau_E) = p(\tau_A)p(\tau_E)$, with $p(\tau) := \nu(s_1)\pi(a_1|s_1) \prod_{t=2}^{T} P_T(s_t|s_{t-1}, a_{t-1})\pi(a_t|s_t)$. In this work, we reformulate GAIL [Ho and Ermon, 2016] in the Bayesian framework as follows.

### 3.1 Bayesian Framework for Adversarial Imitation Learning

**Agent-expert discrimination** Suppose $\pi_A$ is fixed for simplicity, which will be later parameterized for learning. Let us consider binary auxiliary random variables $o_{A,t}, o_{E,t}$ for all $t$, where $o_t$ becomes 1 if given state-action pair $(s_t, a_t)$ is generated by the expert, and becomes 0 otherwise. Then, the

joint distribution of $(\boldsymbol{\tau}_A, \boldsymbol{\tau}_E, \boldsymbol{o}_A, \boldsymbol{o}_E)$ can be written as

$$p(\tau_A, \tau_E, o_A, o_E) = p(\tau_A)p(\tau_E)\left[\prod_{t=1}^{T} p(o_{A,t}|s_{A,t}, a_{E,t})\right]\left[\prod_{t=1}^{T} p(o_{E,t}|s_{A,t}, a_{E,t})\right]. \quad (3)$$

for $\boldsymbol{o} := (\boldsymbol{o}_t)_{t=1}^{T} := (\boldsymbol{o}_1, ..., \boldsymbol{o}_T)$, where $o_t$ is a realization of a random variable $\boldsymbol{o}_t$. Although $p(o_t|s_t, a_t)$ cannot be the same for both agent and expert and all $t$, we can simplify the problem by applying a single approximate discriminator $D_\phi(s, a)$ with parameter $\phi$ such that

$$p(o_t|s_t, a_t) \approx \hat{p}(o_t|s_t, a_t; \phi) := (1 - D_\phi(s_t, a_t))^{o_t} D_\phi(s_t, a_t)^{1-o_t} = \begin{cases} 1 - D_\phi(s_t, a_t), & \text{if } o_t = 1, \\ D_\phi(s_t, a_t), & \text{otherwise.} \end{cases}$$
$$(4)$$

Using the approximation in (4), the distribution in (3) is given by

$$p(\tau_A, \tau_E, o_A, o_E) \approx \hat{p}(\tau_A, \tau_E, o_A, o_E; \phi) \quad (5)$$

$$:= p(\tau_A)p(\tau_E)\prod_{t=1}^{T}\hat{p}(o_{A,t}|s_{A,t}, a_{A,t}; \phi)\prod_{t=1}^{T}\hat{p}(o_{E,t}|s_{E,t}, a_{E,t}; \phi). \quad (6)$$

It should be noted that the distribution in (6) works for the arbitrary choice of $\tau_A, \tau_E, o_A, o_E$. Also, the graphical model for those random variables is shown in Figure 1 to clarify the dependencies between random variables.

Now, suppose a *discrimination optimality event* $\boldsymbol{o}_A = \boldsymbol{0}$, $\boldsymbol{o}_E = \boldsymbol{1}$ is observed for some fixed trajectories $\tau_A, \tau_E$, where $\boldsymbol{1} := (1)_{t=1}^{T} := (1, ..., 1)$ and $\boldsymbol{0}$ is defined in a similar way. Intuitively, the discrimination optimality event is an event such that the discriminator perfectly recognizes the policy that generates given state-action pairs. By introducing a prior $p(\phi)$ on the discriminator parameter $\phi$ and the agent policy $\pi_A(\cdot|\cdot; \theta)$ parameterized with $\theta$, we obtain the following posterior distribution conditioned on the discrimination optimality event and $\theta$:

$$p(\phi, \tau_A, \tau_E|\boldsymbol{0}_A, \boldsymbol{1}_E; \theta) \propto p(\phi)p(\tau_A; \theta)p(\tau_E)p(\boldsymbol{0}_A|\tau_A; \phi)p(\boldsymbol{1}_E|\tau_E; \phi). \quad (7)$$

Here, $\boldsymbol{0}_A$ and $\boldsymbol{1}_E$ is defined as the events $\boldsymbol{o}_A = \boldsymbol{0}$ and $\boldsymbol{o}_E = \boldsymbol{1}$, respectively. By using the posterior $p(\phi|\boldsymbol{0}_A, \boldsymbol{1}_E; \theta)$ which marginalizes out $\tau_A$ and $\tau_E$ in (7), we can consider the full distribution of $\phi$ or select an appropriate point estimate for $\phi$ that maximizes the posterior.

**Discrimination-based imitation** Suppose we want to find the parameter $\theta$ of $\pi_A$ that well approximates $\pi_E$ based on the discrimination results. By considering parameters $(\theta, \phi)$ as random variables $(\boldsymbol{\theta}, \boldsymbol{\phi})$, the distribution for $(\boldsymbol{\tau}_A, \boldsymbol{\tau}_E, \boldsymbol{o}_A, \boldsymbol{o}_E, \boldsymbol{\theta}, \boldsymbol{\phi})$ is

$$p(\tau_A, \tau_E, o_A, o_E, \theta, \phi) = p(\theta)p(\phi)p(\tau_A, \tau_E, o_A, o_E; \theta, \phi) \quad (8)$$

$$= p(\theta)p(\phi)p(\tau_A; \theta)p(\tau_E)\prod_{t=1}^{T}\hat{p}(o_{A,t}|s_{A,t}, a_{A,t}; \phi)\prod_{t=1}^{T}\hat{p}(o_{E,t}|s_{E,t}, a_{E,t}; \phi), \quad (9)$$

where $\boldsymbol{\phi}$ is assumed to be independent with $\boldsymbol{\theta}$. Similar to the optimism for the agent-expert discrimination, suppose we observe the *imitation optimality event* $\boldsymbol{o}_A \neq \boldsymbol{0}$ that is irrespective of $\boldsymbol{o}_E$. Note that the imitation optimality event implies preventing the occurrence of discrimination optimality events. To get the optimal policy parameter by using the discriminator, we can consider the following (conditional) posterior:

$$p(\theta, \tau_A|\tilde{\boldsymbol{0}}_A; \phi) \propto p(\theta)p(\tau_A; \theta)p(\tilde{\boldsymbol{0}}_A|\tau_A; \phi). \quad (10)$$

Here, $\tilde{\boldsymbol{0}}_A$ is defined as an probabilistic event $\boldsymbol{o}_A \neq \boldsymbol{0}$. Finally by using $p(\theta|\tilde{\boldsymbol{0}}_A; \phi)$ that comes from the marginalization of $\tau_A$ in (10), either the full distribution of $\theta$ or corresponding point estimate can be used.

### 3.2 GAIL as an Iterative Point Estimator

Under our Bayesian framework, GAIL can be regarded as an algorithm that iteratively uses (7) and (10) for updating $\theta$ and $\phi$ using their point estimates. For the discriminator update, the objective of

GAIL is to maximize the expected log-likelihood with $\theta_{\text{prev}}$ given from the previous iteration and $\boldsymbol{\tau}_A$ generated by using $\pi_A(a|s; \theta_{\text{prev}})$:

$$\arg\max_{\phi} \mathbb{E}_{\boldsymbol{\tau}_A, \boldsymbol{\tau}_E | \boldsymbol{\theta}=\theta_{\text{prev}}} \left[ \log p(\boldsymbol{0}_A | \boldsymbol{\tau}_A, \phi) p(\boldsymbol{1}_E | \boldsymbol{\tau}_E, \phi) \right] \tag{11}$$

$$= \arg\max_{\phi} \mathbb{E}_{\boldsymbol{\tau}_A, \boldsymbol{\tau}_E | \boldsymbol{\theta}=\theta_{\text{prev}}} \left[ \sum_{t=1}^{T} \log D_{\phi}(\boldsymbol{s}_{A,t}, \boldsymbol{a}_{A,t}) + \sum_{t=1}^{T} \log(1 - D_{\phi}(\boldsymbol{s}_{E,t}, \boldsymbol{a}_{E,t})) \right]. \tag{12}$$

This can be regarded as a surrogate objective with an uninformative prior $p(\phi)$ since

$$\log p(\boldsymbol{0}_A, \boldsymbol{1}_E | \phi, \theta_{\text{prev}}) = \log \mathbb{E}_{\boldsymbol{\tau}_A, \boldsymbol{\tau}_E | \boldsymbol{\theta}=\theta_{\text{prev}}} \left[ p(\boldsymbol{0}_A | \boldsymbol{\tau}_A, \phi) p(\boldsymbol{1}_E | \boldsymbol{\tau}_E, \phi) \right] + constant \tag{13}$$

$$\geq \mathbb{E}_{\boldsymbol{\tau}_A, \boldsymbol{\tau}_E | \boldsymbol{\theta}=\theta_{\text{prev}}} \left[ \log p(\boldsymbol{0}_A | \boldsymbol{\tau}_A, \phi) p(\boldsymbol{1}_E | \boldsymbol{\tau}_E, \phi) \right] + constant, \tag{14}$$

where the inequality in (14) follows from the Jensen's inequality. For the policy update, the objective of GAIL is

$$\arg\max_{\theta} \mathbb{E}_{\boldsymbol{\tau}_A | \boldsymbol{\theta}=\theta} \left[ \log p(\tilde{\boldsymbol{0}}_A | \boldsymbol{\tau}_A, \phi_{\text{prev}}) \right] = \arg\min_{\theta} \mathbb{E}_{\boldsymbol{\tau}_A | \boldsymbol{\theta}=\theta} \left[ \sum_{t=1}^{T} \log D_{\phi_{\text{prev}}}(\boldsymbol{s}_{A,t}, \boldsymbol{a}_{A,t}) \right]. \tag{15}$$

Similarly, for the uninformative prior $p(\theta)$, we can show that

$$\log p(\tilde{\boldsymbol{0}}_A | \theta, \phi_{\text{prev}}) = \log \mathbb{E}_{\boldsymbol{\tau}_A | \boldsymbol{\theta}=\theta} \left[ p(\tilde{\boldsymbol{0}}_A | \boldsymbol{\tau}_A, \phi_{\text{prev}}) \right] + constant \tag{16}$$

$$\geq \mathbb{E}_{\boldsymbol{\tau}_A | \boldsymbol{\theta}=\theta} \left[ \log p(\tilde{\boldsymbol{0}}_A | \boldsymbol{\tau}_A, \phi_{\text{prev}}) \right] + constant, \tag{17}$$

and thus, the objective in (15) can be regarded as a surrogate objective. In addition, since the form of the objective in (15) is the same as the policy optimization with an immediate *cost* function $\log D_{\phi_{\text{prev}}}(\cdot, \cdot)$, GAIL uses TRPO, a state-of-the-art policy gradient algorithm, for updating $\theta$.

Note that our approach shares the same insight behind the probabilistic inference formulation of reinforcement learning, in which the reinforcement learning problem is casted into the probabilistic inference problem by introducing the auxiliary *return optimality event* [Toussaint, 2009, Neumann, 2011, Abdolmaleki et al., 2018]. Also, if we consider the maximization of $\log p(\boldsymbol{1}_A | \theta, \phi_{\text{prev}})$, which result from defining the imitation optimality event as $\boldsymbol{o}_A = \boldsymbol{1}$, it can be shown that the corresponding surrogate objective becomes the policy optimization with an immediate *reward* function $\log(1 - D_{\phi_{\text{prev}}}(\cdot, \cdot))$. This is in line with speeding up GAN training by either maximizing $\log(1 - D(\cdot))$ or minimizing $\log D(\cdot)$, suggested in Goodfellow et al. [2014]. Some recent work on adversarial inverse reinforcement learning also support the use of such reward function [Finn et al., 2016, Fu et al., 2018].

### 3.3 Sample-efficient Imitation Learning with Predictive Cost Function

Since model-free imitation learning algorithms (e.g. GAIL) require experience samples obtained from the environment, improving the sample-efficiency is critical. From the Bayesian formulation in the previous section, GAIL can be seen as maximizing (minimizing) the expected log-likelihood in a point-wise manner for discriminator (policy) updates, and this makes the algorithm quite inefficient compared to using the full predictive distribution.

We thus propose to use the posterior of the discriminator parameter so that more robust cost signals are available for policy training. Formally, let us consider the iterative updates for the policy parameter $\theta$ and the discriminator parameter $\phi$, where the point estimate of $\theta$ is obtained using the distribution over $\phi$ in each iteration. In other words, given $\theta_{\text{prev}}$ from the previous iteration, we want to utilize $p_{\text{posterior}}(\phi) := p(\phi | \boldsymbol{0}_A, \boldsymbol{1}_E, \theta_{\text{prev}})$ that satisfies

$$\log p_{\text{posterior}}(\phi) = \log \left\{ p(\phi) \mathbb{E}_{\boldsymbol{\tau}_A | \boldsymbol{\theta}=\theta_{\text{prev}}} \left[ p(\boldsymbol{0}_A | \boldsymbol{\tau}_A, \phi) \right] \mathbb{E}_{\boldsymbol{\tau}_E} \left[ p(\boldsymbol{1}_E | \boldsymbol{\tau}_E, \phi) \right] \right\} + constant. \tag{18}$$

By using Monte-Carlo estimations for the expectations over trajectories in (18), the log posterior in (18) can be approximated as

$$\log p(\phi) + \log \sum_{n=1}^{N} \exp(F_{A,\phi}^{(n)}) + \log \sum_{n=1}^{N} \exp(F_{E,\phi}^{(n)}) + constant, \tag{19}$$

where $F_{A,\phi}^{(n)} := \sum_{t=1}^T \log D_\phi(s_{A,t}^{(n)}, a_{A,t}^{(n)})$ and $F_{E,\phi}^{(n)} := \sum_{t=1}^T \log(1 - D_\phi(s_{E,t}^{(n)}, a_{E,t}^{(n)}))$. Note that we can also use the surrogate objective of GAIL in (14) with prior on $p(\phi)$, which might be suitable for the infinite-horizon problems.

At each iteration of our algorithm, we try to find policy parameter $\theta$ that maximizes the log of the posterior $\log p(\theta | \tilde{0}_A)$. For an uninformative prior on $\theta$, the objective can be written as

$$\arg \max_\theta \log p(\theta | \tilde{0}_A) = \arg \min_\theta \log p(0_A | \theta) = \arg \min_\theta \log \mathbb{E}_{\boldsymbol{\tau}_A | \boldsymbol{\theta} = \theta, \boldsymbol{\phi} \sim p_{\text{posterior}}}[p(0_A | \boldsymbol{\tau}_A, \boldsymbol{\phi})]. \tag{20}$$

By applying the Jensen's inequality to (20), we have $\mathbb{E}_{\boldsymbol{\tau}_A | \boldsymbol{\theta} = \theta, \boldsymbol{\phi} \sim p_{\text{posterior}}}[\log p(0_A | \boldsymbol{\tau}_A, \boldsymbol{\phi})]$, which can be minimized by policy optimization. In contrast to GAIL that uses the single point estimate for the maximization of $p_{\text{posterior}}$, multiple parameters $\phi_1, ..., \phi_K$ that are randomly sampled from $p_{\text{posterior}}$ are used to estimate the objective:

$$\mathbb{E}_{\boldsymbol{\tau}_A | \boldsymbol{\theta} = \theta} \left\{ \frac{1}{K} \sum_{k=1}^K \log p(0_A | \boldsymbol{\tau}_A, \phi_k) \right\} = \mathbb{E}_{\boldsymbol{\tau}_A | \boldsymbol{\theta} = \theta} \left\{ \frac{1}{K} \sum_{k=1}^K \sum_{t=1}^T \log D_{\phi_k}(\boldsymbol{s}_{A,t}, \boldsymbol{a}_{A,t}) \right\} \tag{21}$$

$$= \mathbb{E}_{\boldsymbol{\tau}_A | \boldsymbol{\theta} = \theta} \left\{ \sum_{t=1}^T \left( \frac{1}{K} \sum_{k=1}^K \log D_{\phi_k}(\boldsymbol{s}_{A,t}, \boldsymbol{a}_{A,t}) \right) \right\}. \tag{22}$$

Note that (22) implies we can perform RL policy optimization with the *predictive* cost function $\frac{1}{K} \sum_{k=1}^K \log D_{\phi_k}(s, a)$. In addition, if we consider $p(1_A | \boldsymbol{\tau}_A, \phi_k)$ rather than $p(\tilde{0}_A | \boldsymbol{\tau}_A, \phi_k)$, the optimization problem becomes RL with the predictive reward function $\frac{1}{K} \sum_{k=1}^K (1 - \log D_{\phi_k}(s, a))$. The remaining question is how to get the samples from the posterior, and this will be discussed in the next section.

## 4 Posterior Sampling Based on Stein Variational Gradient Descent (SVGD)

SVGD [Liu and Wang, 2016] is a recently proposed Bayesian inference algorithm based on the particle updates, which we briefly review as follows: suppose that random variable $\boldsymbol{x}$ follows the distribution $q^{(0)}$, and target distribution $p$ is known up to the normalization constant. Also, consider a sequence of transformations $T^{(0)}, T^{(1)}, ...$, where

$$T^{(i)}(x) := x + \epsilon^{(i)} \psi_{q^{(i)}, p}(x), \quad \psi_{q,p}(x') := \mathbb{E}_{\boldsymbol{x} \sim q}[k(\boldsymbol{x}, x') \nabla_{\boldsymbol{x}} \log p(\boldsymbol{x}) + \nabla_{\boldsymbol{x}} k(\boldsymbol{x}, x')] \tag{23}$$

with sufficiently small step size $\epsilon^{(i)}$, probability distribution $q^{(i)}$ of $(T^{(i-1)} \circ \cdots T^{(0)})(\boldsymbol{x})$ and some positive definite kernel $k(\cdot, \cdot)$. Interestingly, the *deterministic* transformation (23) turns out to be an iterative update to the probability distribution towards the target distribution $p$, and $\psi_{q^{(i)}, p}$ can be interpreted as the functional gradient in the reproducing kernel Hilbert space (RKHS) defined by the kernel $k(\cdot, \cdot)$. SVGD was shown to minimize the kernelized Stein discrepancy $\mathbb{S}(q^{(i)}, p)$ between $q^{(i)}$ and $p$ [Liu et al., 2016] in each iteration. In practice, SVGD uses a finite number of particles. More formally, for $K$ particles $\{x_k^{(0)}\}_{k=1}^K$ that are initially sampled, SVGD iteratively updates those particles by the following transformation that approximates (23):

$$T^{(i)}(x) := x + \epsilon^{(i)} \hat{\psi}_p^{(i)}(x), \quad \hat{\psi}_p^{(i)}(x) := \frac{1}{K} \sum_{k=1}^K \left( k(x_k^{(i)}, x) \nabla_{x_k^{(i)}} \log p(x_k^{(i)}) + \nabla_{x_k^{(i)}} k(x_k^{(i)}, x) \right). \tag{24}$$

Even with the approximate deterministic transform and a few particles, SVGD was experimentally shown to significantly outperform common Bayesian inference algorithms. In the extreme case, if a single particle is used, SVGD is equivalent to MAP inference.

In our work, we use SVGD to draw the samples of the discriminator parameters from the posterior (19). Specifically, we first choose a set of $K$ initial particles (discriminator parameters) $\{\phi_k^{(0)}\}_{k=1}^K$. Then, we use the gradient of (19) for those particles and apply the update rule in (24) to get the particles generated from the posterior distribution in (19). Finally, by using those particles, the predictive cost function is derived. The complete BGAIL algorithm leveraging SVGD and the predictive cost function is summarized in **Algorithm 1**.

---

**Algorithm 1** Bayesian Generative Adversarial Imitation Learning (BGAIL)

---

1: **Input:** Expert trajectories $\mathcal{D}_E$, initial policy parameter $\theta$, a set of initial discriminator parameters $\{\phi_k\}_{k=1}^K$, $p(\phi)$ for preference of $\phi$
2: **for each iteration do**
3:     Sample trajectories by using policy $\pi_\theta$.
4:     Update $\theta$ using policy optimization, e.g., TRPO, with cost function $\frac{1}{K}\sum_{k=1}^K \log D_{\phi_k^{(i)}}(s,a)$.
5:     Sample trajectories from $\mathcal{D}_E$.
6:     **for** $k = 1, ..., K$ **do**
7:         Calculate gradient $\delta_k$ of either (19) or its surrogate objective (17) for $\phi_k$.
8:     **end for**
9:     **for** $k = 1, ..., K$ **do**                                              ▷ SVGD
10:         Update $\phi_k \leftarrow \phi_k + \alpha\hat{\psi}(\phi_k)$ for a step size parameter $\alpha$, where

$$\hat{\psi}(\phi) := \frac{1}{K}\sum_{j=1}^K \left( k(\phi_j, \phi)\delta_j + \nabla_{\phi_j} k(\phi_j, \phi) \right).$$

11:     **end for**
12: **end for**

---

## 5   Experiments

We evaluated our BGAIL on five continuous control tasks (Hopper-v1, Walker2d-v1, HalfCheetah-v1, Ant-v1, Humanoid-v1) from OpenAI Gym, implemented with the MuJoCo physics simulator [Todorov et al., 2012]. We summarize our experimental setting as follows. For all tasks, neural networks with 2 hidden layers were used for all policy and discriminator networks, where 100 hidden units for each hidden layer and $\tanh$ activations are used. Before training, expert's trajectories were collected from the expert policy released by the authors of the original GAIL[1], but our code was built on the GAIL implementation in OpenAI Baselines [Dhariwal et al., 2017] which uses Tensor-Flow [Abadi et al., 2016]. For the policy, Gaussian policy was used with both mean and variance dependent on the observation. For the discriminator, the number of particles $K$ was chosen to be 5. All discriminator parameters $\phi_1, ..., \phi_K$ were initialized independently and randomly. For training, we used uninformative prior and SVGD along with the Adam optimizer [Kingma and Ba, 2014], whereas Adagrad was used in the SVGD paper [Liu and Wang, 2016]. Our SVGD was implemented using the code released by the authors[2], with the radial basis function (RBF) kernel (squared-exponential kernel) $k(\cdot, \cdot)$ and the median heuristic for choosing the bandwidth parameter. In addition, 5 inner loops were used for updating discriminator parameters, which corresponds to the inner loop from line 6 to line 11 in **Algorithm 1**.

First, we compare BGAIL to two different settings for GAIL. The first setting is the same as in the authors' code, where the variance of the Gaussian policy is learnable constant parameter and a single discriminator update is performed in each iteration. Also, the state-action pairs of the expert demonstration were subsampled from complete trajectories. In the second setting, we made changes to the original setting to improve sample efficiency by (1) state-dependent variance and (2) 5 discriminator updates per iteration, and (3) use the whole trajectories without sub-sampling. In the remainder of this paper, these two settings shall be referred to as vanilla GAIL and tuned GAIL, respectively. In all settings of our experiments, the maximum number of expert trajectories was chosen as in Ho and Ermon [2016], i.e. 240 for Humanoid and 25 for all other tasks, and 50000 state-action pairs were used for each iteration in the first experiment. The number of training iterations were also also chosen as the same as written in GAIL paper. The imitation performances of vanilla GAIL, tuned GAIL and our algorithm are summarized in Table 1. Note that the evaluation in Table 1 was done in the exactly same setting as the original GAIL paper. In that paper, the imitation learner was evaluated over 50 independent trajectories using the single trained policy, and the mean and the standard deviation of those 50 trajectories were given. Similarly, we evaluated each of the 5 trained policies over 50 independent trajectories, and we reported the mean and the standard deviation over 50 trajectories of the 3rd best policy in terms of the mean score for fair comparison. As we can see, tuned

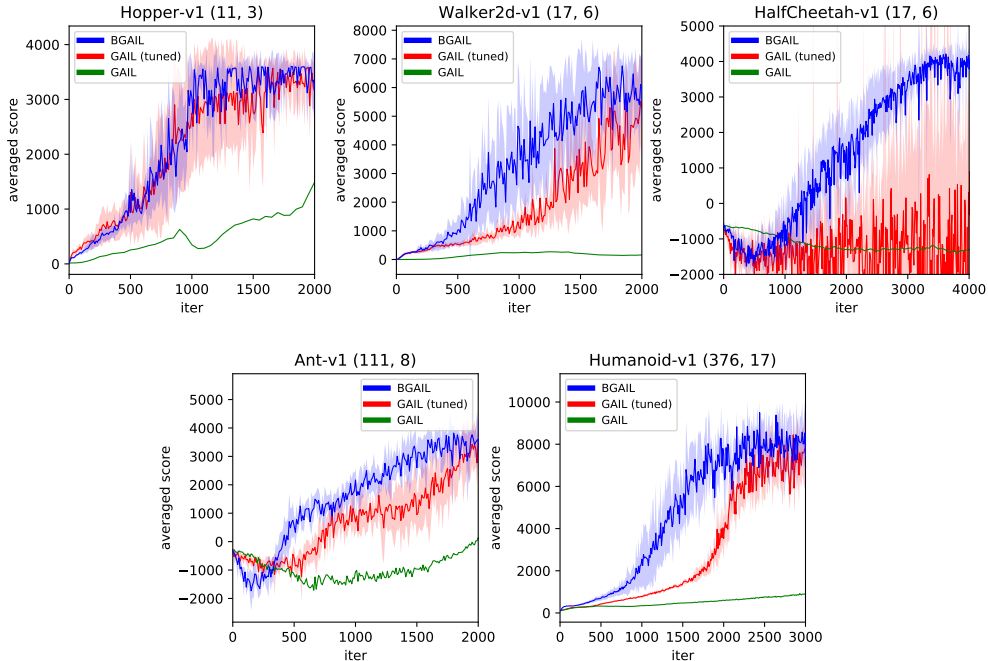

Figure 2: Comparison with GAIL when either 1000 (Hopper-v1, Walker-v1, HalfCheetah-v1) or 5000 (Ant-v1, Humanoid-v1) state-action pairs are used for each training iteration. The numbers inside the bracket on the titles indicate (from left to right) the state dimension and the action dimension of the task, respectively. The tasks are ordered by following the ascending order of the state dimension.

| Task | Dataset size | GAIL (released) | GAIL (tuned) | BGAIL |
|---|---|---|---|---|
| Hopper-v1 | 25 | $3560.85 \pm 3.09$ | $3595.30 \pm 5.89$ | $\mathbf{3613.94 \pm 10.25}$ |
| Walker2d-v1 | 25 | $6832.01 \pm 254.64$ | $7011.02 \pm 25.18$ | $\mathbf{7017}.46 \pm \mathbf{33.32}$ |
| HalfCheetah-v1 | 25 | $4840.07 \pm 95.36$ | $\mathbf{5022.93 \pm 81.46}$ | $4970.77 \pm 363.48$ |
| Ant-v1 | 25 | $4132.90 \pm 878.67$ | $4759.12 \pm 416.15$ | $\mathbf{4808.90 \pm 78.10}$ |
| Humanoid-v1 | 240 | $10361.94 \pm 61.28$ | $10329.66 \pm 59.37$ | $\mathbf{10388.34 \pm 99.03}$ |

Table 1: Imitation Performances for vanilla GAIL, tuned GAIL and BGAIL

GAIL and BGAIL perform slightly better than vanilla GAIL for most of the tasks and hugely better at Ant-v1. We think this is due to (1) the expressive power by using the policy with state-dependent variance, (2) the stabilization of the algorithm due to the multiple iteration for discriminator training and (3) the efficient use of expert's trajectories without sub-sampling procedure.

Second, we checked the sample efficiency of our algorithm by reducing the number of state-action pairs used for each training iteration from 50000 to 1000 for Hopper-v1, Walker2d-v1, HalfCheetah-v1 and to 5000 for other much high-dimensional tasks. Note that the vanilla GAIL in this experiment used 50000 state-action pairs to see the sample efficiency of the original work, whereas the tuned GAIL was trained with either 1000 or 5000 state-action pairs per each iteration to compare its sample efficiency with our algorithm. Compared to vanilla GAIL, the performances of both tuned GAIL and BGAIL converge to the optimal (expert's performance) much faster as depicted in Figure 2. Note that 5 different policies were trained for both BGAIL and tuned GAIL, whereas a single policy was trained for vanilla GAIL. The shades in Figure 2 indicate the standard deviation of scores over these 5 policies. Also, it can be shown that the performances of tuned GAIL and BGAIL are almost similar at Hopper-v1 that is relatively a low-dimensional task. On the other hand, as the dimension of tasks increases, BGAIL becomes much more sample-efficient compared to tuned GAIL.

# 6 Discussion

In this work, GAIL is analyzed in the Bayesian approach, and such approach can lead to highly sample-efficient model-free imitation learning. Our Bayesian approach is related to Bayesian GAN [Saatci and Wilson, 2017] that considered the posterior distributions of both generator and discriminator parameters in the generative adversarial training. Similarly in our work, the posterior for the agent-expert discriminator was used for the predictive density of the cost during training, whereas only a point estimate for the policy parameter was used for simplicity. We think our algorithm can be simply extended to the multi-policy imitation learning, and the sample efficiency of our algorithm may be enhanced by utilizing the posterior of the policy parameter as shown in Stein variational policy gradient (SVPG) [Liu et al., 2017]. Also for the theoretical analysis, ours slightly differs from the analysis in Bayesian GAN due to the inter-trajectory correlation from MDP formulation in our work. This makes the objective of original GAIL regarded as the surrogate objective in our Bayesian approach, whereas the objective of Bayesian GAN is exactly reduced into that of original GAN for ML point estimation. In addition, we think our analysis fills the gap between theory and experiments in GAIL since GAIL was theoretically analyzed based on the discounted occupancy measure, which can be defined in the infinite-horizon setting, but their experiments were only done on the finite-horizon tasks in MuJoCo simulator. Finally, while BGAIL effectively works with uninformative prior in our experiments, the proper choice of the prior such as Gaussian prior with Fisher information covariance in [Abdolmaleki et al., 2018]. may also enhance the sample efficiency.

## Acknowledgement

This work was supported by the ICT R&D program of MSIT/IITP. (No. 2017-0-01778, Development of Explainable Human-level Deep Machine Learning Inference Framework) and the Ministry of Trade, Industry & Energy (MOTIE, Korea) under Industrial Technology Innovation Program (No.10063424, Development of Distant Speech Recognition and Multi-task Dialog Processing Technologies for In-door Conversational Robots).

## Footnotes

[1]https://github.com/openai/imitation

[2]https://github.com/DartML/Stein-Variational-Gradient-Descent

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
