[Supplementary Material · supplementary.pdf]

# Appendix A    Additional Experiments

**More performance evaluation experiments.** We compare BGAIL with (tuned) GAIL by varying the batch size (1000, 3000, 5000, 10000) in each iteration. For both BGAIL and GAIL, we train 5 policies for 500 iterations (Walker2d, HalfCheetah) and 1000 iterations (Ant), evaluate the policies over 50 trajectories and pick the 3rd best policy in terms of the mean score. The results are summarized in Figure 3, where the x-axis is the batch size: BGAIL performs significantly better than GAIL when the batch size is small.

**Computation time of using different number of particles.** We report the computation times of BGAIL under different numbers of particles in Figure 4. Computation times are measured for a single iteration on Intel Xeon E5-2620 (16 cores) with a single GPU GTX1080. The computation times are averaged over the first 20 iterations in the training, and the batch size is fixed to 1000 for all tasks. Our implementation uses MPI to parallelize the computations by exploiting the decomposability of SVGD particles, thus the running time scales linearly in the number of particles.

**Performance of using different number of particles.** Figure 5 compares the performance of BGAIL under varying number of particles (1, 5, 9) in Walker2d. We show the moving averages of the mean score over 15 learning trials. We observe that using more particles improves performance.

Figure 3: Performance evaluation for different batch sizes

Figure 4: Computation time.

Figure 5: Performance of using multiple particles.

# Appendix B   Hyperparameter settings

| Hyperparameters | Values |
|---|---|
| maximum KL divergence (TRPO) | 0.01 |
| iterations for conjugate gradient (CG) | 10 |
| $\gamma$ (discount factor) | 0.995 |
| $\lambda$ (GAE) | 0.97 |
| entropy coefficient (TRPO) | 0.0 |
| CG damping factor | 0.1 |
| stepsize for value function | 0.001 |
| iterations for value function updates | 5 |
| entropy coefficient for discriminator | 0.001 |
| stepsize for discriminator updates | 0.01 |
| normalization for observations | true |