[Reviews · NeurIPS 2018]

Reviewer 1



The authors propose a Bayesian formulation of Generative Adversarial Imitation Learning (GAIL) and demonstrate that this leads to better sample efficiency in terms of environment interaction. Contributions: 1. The authors show that GAIL can be viewed as a iterative maximum likelihood point estimation method. 2. The authors introduce a Bayesian formulation of GAIL (BGAIL) that uses posterior sampling. 3. The authors demonstrate the sample efficiency of BGAIL on simulated domains with high-dimensional states and actions. The contributions in the paper are clear. The paper is well-written, and derivations are presented in a straightforward manner. The result for the HalfCheetah task in figure 1 are rather curious (why does GAIL perform so poorly?). It would be useful to comment on this. However, in general, it does seem that BGAIL results in fewer environment interactions for policy learning. That said, the experimental evaluation is a key aspect of the claims of the paper. The formalism to show the Bayesian perspective would leave a reader convinced if experimental evaluation was a bit more thorough. This would mean performing experiments in a number of environments (even if in sim) to show that any gains achieved by BGAIL are not due to some dynamics characterstic in mujoco continuous control suite, but indeed due to the algorithm.

Reviewer 2



SUMMARY The paper proposes an improvement to the Generative Adversarial Imitation Learning (GAIL) approach. In GAIL, a discriminator (critic) function is updated to provide feedback on how close the distribution induced by the policy is to the expert distribution. The generator (policy) is then updated to make these distributions more similar. These updates happen interatively. The paper proposes to make this procedure more robust by defining a *distribution* over discriminator parameters, from which multiple samples are drawn. The generator then tries to improve wrt all of these samples, rather than a point estimate. TECHNICAL QUALITY The paper uses mostly experiments (rather than, say, theoretical proofs) to support the claimed contributions. The new method is compared to two relevant baselines - published GAIL and a 'tuned GAIL' that uses the proposed innovations except the use of multiple posterior samples. Experiments are over five different common benchmarks for imitation and reinforcement learning, using as much as possible standardised settings from earlier papers. It is not clear how many trials were performed for each setting - and thus how reliable the averages are. It seems single trials were used in each environment. (but at least the findings were relatively consistent over the five tasks). The development of the method seems mostly correct, I found the following (relatively minor) issues: - paper states ML is a surrogate objective for a bayesian approach. It could be regarded as surrogate objective for a MAP estimate, but in a full Bayesian approach p(\phi|...) or p(\theta|...) aren't objectives, but the sought quantity (i.e., one doesn't maximise the posterior). - The main reason for the efficiency of BGAIL seems to be that using multiple samples of the posterior over discriminators (cost functions) seems more robust than using a point estimate. It would be interesting to describe this point in more detail: why is using a point estimate so brittle? Is there an experiment that could be done to get more insight in this point? - The paper implies (23) as another estimate for the quantity in (17), but in fact it is a different objective altogether (I.e. doesn't share LHS with (17)). That could be clearer. - It seems that BGAIL would be more costly computationally because of the SVGD procedure. (as a tradeoff for a higher sample efficiency). Could you quantify the difference in computation time? - It wasn't clear to me what the +- indicates in table 1. Standard deviation over independent trials (how many)? Or st. dev within a trial? Or standard error, or multiple deviations, etc. Interpreting them as, say, 95 % confidence interval, most differences do not seem statistically significant between tuned GAIL and BGAIL in table 1 (but besides this asymptotic performance, learning seems to be a lot quicker for BGAIL) in the shown run. That makes it hard to judge the impact on asymptotic performance. RELEVANCE, SIGNIFICANCE, NOVELTY The imitation learning problem is a relevant problem to the machine learning community, as a problem in itself and as a promising way to initialise reinforcement learning policies. The approach seems to be more sample efficient then tuned GAIL (the most competitive baseline) - although the effect depends a lot on the environment, with relatively modest improvements on some of the environments. Asymptotic performance is roughly the same. I would thus judge the work as somewhat significant. (I think the impact could increase if a bit more focused was put on the problem with brittleness and how that is solved by the BGAIL approach). The approach is somewhat novel, although it is still quite similar to the conventional GAIL approach. The derivation it self is interesting, although in some part it isn't clear whether a full Bayesian approach or using MAP estimates / posterior means are meant. CLARITY The writing is mostly clear and well-structured. At times, there are small grammar issues (e.g. a missing article in "related to variational approach"). At two points, I felt the clarity of the text could be improved: -In 2.1 RL machinery is introduced - maybe it would be better to explain this step: we want to learn a cost function such that the gibbs policy follows the expert distribution as well as possible. For that, we need, Q, V, etc Derivation around the introduction of the 'discrimination events' is a bit meandering. - The discussion in paragraph 3.1 feels a bit convoluted with the introduction of the variables o_A, o_E, values 0_A, 1_E etcetera. It seems that this could perhaps be expressed more concisely using the output of the discriminator (and the true label) as functions, rather than introducing new random variables. Further, it seems the algorithm is described in sufficient detail to be re-implemented. The experiments are missing some detail to be reproduced or interpreted (e.g. how can the +- in table 1 be interpreted, how many independent training runs were performed for each method). Without knowing the number of independent training runs, it is hard to judge the confidence in obtaining the same result in a reproduction study. MINOR COMMENTS - Line 127: In general, we can't obtain MLE from posterior - The notation with C as wildcard for {A or E} is a bit confusing. It makes it seems like C is a variable and A and E are values. - Line 162: "Imitation learning requires experience samples"-> not always, for example in the case of behaviour cloning where the learned policy is never run - There are many types of RBF kernel, please specify which one you used. (line 218) I have read the rebuttal, and appreciate the extra results as well as the clarification regarding the experimental procedure and the number of policy runs performed with different random seeds. I have updated my score accordingly.

Reviewer 3



Imitation learning in robotics faces several challenges, including the ill-posedness of IRL, as well as sample efficiency. Building on Generative Adversarial Imitation Learning (GAIL), this paper addresses these issues by placing GAIL in a Bayesian framework (BGAIL). Like GAIL, BGAIL does not require an MDP solver in the inner loop, and can learn from a significantly smaller number of samples than GAIL due to the superiority of using an expressive posterior over discriminators, rather than point estimate. Strengths: + Imitation learning generalizes much better than behavioral cloning, but is currently under-utilized in robotics due to data efficiency issues. Thus, this is a highly relevant problem. + The Bayesian setting is well-motivated, both from a theoretical standpoint and evidence from prior work such as Bayesian GAN. The theory for implementing this is not just gluing together BIRL and GAIL, but is quite technical and seeming sound. + It is satisfying to see that BGAIL was derived in such a way that GAIL is a special-case in which point estimates are used, rather than a different algorithm entirely. + The experiments are high-quality. It is nice to see that the paper compares BGAIL not only to GAIL, but to a more highly tuned version of GAIL that makes the comparison more fair. Although BGAIL’s asymptotic improvement over GAIL is very minor, the efficiency gains in the low-data setting (Fig 1) are quite nice and make this a worthwhile improvement that may make imitation learning on robots significantly more tractable. Weaknesses: - It would have been nice to see experiments that showed the effect of using different numbers of particles. It is currently unclear how important this choice is. - The writing is occasionally uneven and could use additional editing (more from a language-usage and type perspective than a technical perspective). Also, in the equations, the notation is exact, but gets quite unwieldy at times. It may be worth considering dropping some of the sub/superscripts for convenience where possible (and noting that you are doing so). - This is a minor detail, but in section 2.2 when discussing the ill-posedness of IRL, the paper focuses on degenerate solutions. But the main ill-posedness problem comes more from the infinite number of reward functions that can describe identical policies / equally explain demonstrations. Overall, the author response addressed several of the concerns of myself and the other reviews, reinforcing my positive outlook on this work.